## Replication

psychology/cognition

replication, Social Value Orientation,
attachment security, reliability of measurements

**Author for correspondence:**
Hans IJzerman
e-mail: h.ijzerman@gmail.com

# Social value orientation and attachment: a replication and extension of Van Lange *et al.* (1997)

## Hans IJzerman[1] and Jaap J. A. Denissen[2]

[1]LIP/PC2S, Université Grenoble Alpes, Grenoble, France
[2]Tilburg University, The Netherlands

HIJ, 0000-0002-0990-2276

We report a replication and extension of a finding from Studies 1 and 2 of Van Lange *et al.*'s influential paper (Van Lange *et al.* 1997 *J. Pers. Soc. Psychol.* **73**, 733–746. (doi:10.1037/0022-3514.73.4.733)), which showed an association between Social Value Orientation (SVO) and attachment security. We report a close replication but with measures of attachment that are considered superior in comparison to measures used by Van Lange *et al.*, due to subsequent psychometric improvements. Psychometric analyses indeed showed that our attachment measures were reliable and valid, demonstrating theoretically predicted associations with other outcomes. With a sample ($N = 879$) sufficiently large to detect $d = 0.19$ (and larger than the original $N = 573$), we failed to replicate the effect. Based on the available evidence, we interpret as there being no evidence for the link between attachment security and Social Value Orientation, but further replication research that uses solid measures and large samples can provide more definite conclusions about the association between attachment and SVO.

## 1. Introduction

Social Value Orientation (SVO; [1]) pertains to the value that individuals place on the outcomes of other people, including strangers. In social psychology, an impressive line of research has been devoted to individual differences in SVO, which is a continuum that describes inclinations of some people 'to give interdependent others the benefit of the doubt and approach them cooperatively, whereas other people are inclined to approach interdependent others in a less cooperative manner' ([1, p. 733]; see also [2,3]). People are often divided along these SVOs into three orientations: prosocial (those who maximize their outcome for self and others while minimizing differences

between self and others), individualist (those who maximize their own outcomes without regard for others' outcomes) and competitive (those who maximize their own outcomes relative to others' outcomes).

In a landmark paper, Van Lange et al. [1] speculated about a possible developmental origin of this orientation and reported first evidence that people who are more trusting in dyadic relations (i.e. securely attached individuals) were also more prosocial in their behaviours towards generalized others. This finding was groundbreaking as it extended established individual difference patterns with close others (often a carer) to more generalized relationships with strangers. Van Lange and colleagues [1] concluded that early/prior attachment behaviours play a formative and direct role in our behaviour towards strangers. The paper has been central to the field of social dilemmas and social interactions more generally (the paper was cited by 1062 in Google Scholar at the time we wrote this paper) and remains an important resource to date (cited 112 times in 2016 and 90 times in 2017). Despite its importance, we know of no further empirical demonstrations of the relation between SVO and attachment. A follow-up is therefore vital. We provide an important first follow-up through a close replication.

## 1.1. Attachment theory

In the following, we describe the reasoning behind a possible association between attachment and SVO, starting from a conceptual description of attachment. Close, intimate relations are thought to be rooted in an attachment system that helped to solve basic evolutionary pressures related to survival, caregiving and procreation. Central to attachment behaviours is the regulation of proximity between carers and infants [4]. For example, securely attached infants keep close to carers through behaviours like clinging, crying and smiling, in service of keeping temperature stable and environment safe, all the while providing food [5,6].

One of the hallmark premises of attachment theory is that having a responsive and reliable carer (or not) causes greater (or lesser) confidence in others, forming the basis for how people interact later in life ([7,8]; but see [9]). Resulting individual differences (so-called attachment styles) have been suggested to demonstrate considerable stability ([10]; but see also [11–13]). Attachment styles are therefore thought to form the basis for how individuals deal with others throughout their lives [7,14].

In terms of dimensionality, attachment styles have been conceptualized and measured using both a categorical and a dimensional approach. While the categorical approach assumes the existence of three distinct prototypes of attachment (secure, avoidant and anxious-ambivalent), modern dimensional approaches treat attachment style as consisting of two underlying dimensions: anxiety and avoidance. Note that in the latter framework, security is not a separate dimension but a combination of low levels of both anxiety and avoidance (i.e. attachment security has been found to be multidimensional).

## 1.2. Attachment theory and Social Value Orientation

As stated above, adult attachment is seen as the result of prior experiences with carers. Although the majority of attachment research has been conducted in the context of close relationships, some research indeed suggests that people who are securely attached are also more supportive towards strangers [15]. Van Lange and colleagues [1] thus asked whether '[social value orientations] could be, at least in part, a product of early social experience?' [1, p. 733]. To address this question, they focused on correlations with self-reported attachment patterns (Studies 1 and 2) and with having grown up with siblings (Study 3). In the context of this replication, we focus on the former aspect (another replication that was recently conducted focused on having grown up with siblings; [16]).

Van Lange et al. [1] found that prosocials scored higher on measures of attachment security across both studies. In Study 1, results indicated that people who were more prosocial (as compared to competitors and individualists grouped together; hereafter referred to as 'proselfs') scored higher on this scale (i.e. security), but only marginally significantly lower on the scales measuring attachment anxiety and avoidance in their attachment towards a variety of close others. They continued to examine contrasts between prosocials versus competitors and individualist in terms of the latter two dimensions and found that these were statistically significant and supportive of their hypothesis. Moreover, in Study 2 they found that prosocials more strongly endorsed the secure prototype.

## 2. Problems with the original study

Although the findings garnered considerable interest, they had considerable problems. First and foremost, Van Lange *et al.* measured attachment in two ways. In Study 1 ($N = 573$), they used a multi-item questionnaire that seems a precursor of 13 items published by Carnelley and Janoff-Bulman [17] that eventually formed the Relationship Style Questionnaire [18]. This measure tapped into three attachment dimensions, including security. As stated above, this is not consistent with the contemporary understanding of attachment security as a combination of two underlying dimensions.

The lack of dimensionality might be one of the reasons for the low reliability reported for this scale in the Van Lange *et al.* [1] paper. In Study 2 ($N = 136$), they measured attachment styles using a Likert-scale endorsement of each of the three attachment prototypes. Indeed, the reliabilities of Van Lange *et al.*'s [1] Study 1's only attachment measure providing a significant result was low, introducing abundant measurement error that might have contributed to a lack of robustness (depending on which items were included, the alpha ranged between 0.46 and 0.57). The other scales in Study 1 (anxiety and avoidance, alphas between 0.66 and 0.68) produced only marginal differences ($p = 0.09$ and $p = 0.06$, respectively) in a sample of 573 participants. These marginal differences were obtained while controlling for gender *and* after dropping items from the attachment scale related to feelings about one's partner, yet in Study 2 the effect was present for partner-specific attachment.

In terms of validity, there might also have been problems with the attachment measure's translation. In Van Lange *et al.*'s [1] paper, the authors reported that they asked participants about such items as 'I find it easy to trust others,' 'I find it easy to get close to others', and 'I feel comfortable having other people depend on me', 'I am nervous when anyone gets too close' and 'I find that other people don't want to get as close as I would like'. We did not find these items to be representative of their actual measurement, which we obtained from the first author of the Van Lange *et al.* [1] paper (see our appendix A). Attachment items related to reliance on others (a central characteristic of being attached) were dropped, and the items that were retained seemed to rely on a reciprocal exchange of feelings (e.g. 'I do not mind at all to become very personal (e.g. to exchange feelings) with others'; for the full list, see appendix A), rather than a reliance or dependence on others that stresses the 'merging of self and other'.

Van Lange *et al.* [1] thus reported important findings, as they suggest that secure attachment styles generalize to greater prosocial behaviours towards strangers. The paper formed a 'proof-of-concept' of the SVO measure, as it suggested SVO to be rooted in attachment styles. However, despite its continued importance and the issues related to their measurement, these findings have either not been replicated or no replications have been reported in the 20 years after publication. In this report, we describe the results of four studies, in which we tried to replicate the link between SVO and attachment.

## 3. Description of our replication study

The data was collected in four large-scale surveys at the beginning of four consecutive academic years (2011, 2012, 2013, 2014) among Tilburg University's first year bachelor students. The study was thus not planned as a pure replication, which partly explains why we did not implement Van Lange *et al.*'s [1] original attachment measure (the more major explanation being that the original measure was outdated compared to current standards). We first report our sample composition, followed by the reliabilities of each of our scales (per study). We also analysed some other scales that were collected (for which we provide auxiliary analyses on our Open Science Framework project page). In order to provide the original report with maximum chances of being replicated, we merged the datasets where possible (but report per study in our footnotes).

## 4. Method

### 4.1. Participants

We report the descriptives in table 1. Our participant sample relied on a convenience sample and depended on the number of students participating in our test-week. We did thus not do an *a priori* power analysis. A *post hoc* sensitivity analysis in G*Power [19] with 499 prosocials and 269 proselfs, 80% power, alpha = 0.05 (one-tailed) suggested, however, that we could detect effect sizes as small as

**Table 1.** Sex and age of participants in the 2011, 2012, 2013 and 2014 test-weeks.

| year | sex (female/male) | total *N* | age *M* (s.d.) |
|------|-------------------|-----------|----------------|
| 2011 | 215/45 | 260 | 19.46 (3.07) |
| 2012 | 136/30 | 166 | 19.23 (3.17) |
| 2013 | 166/47 | 241 | 19.55 (2.14) |
| 2014 | 145/45 | 196 | 19.48 (2.10) |

$d = 0.19$. In order to test as conservatively as possible, we excluded eight participants who had four duplicate subject IDs. We ran no further studies to test the effect. All participants were first year bachelor students who participated in exchange for course credit. All data and syntaxes needed for these attachment analyses are available at https://osf.io/6kqzy/.

## 4.2. Procedure and measurements

### 4.2.1. Social Value Orientation

The exact composition of the battery of questionnaires in our test-weeks varied throughout the four years. In each year SVO was assessed by the same nine-item triple dominance measure used in the original ([20]; see also [1]). Although reliability estimates are typically not reported for this measure, we converted all nine responses to a binomial format, with 1 corresponding to a prosocial response and 0 to a proself response. Using the R package DescTools [21], we thereafter computed the Kuder–Richardson 20 coefficient as an indicator of reliability. The resulting coefficient was 0.96, indicating a highly consistent pattern of responding either in a prosocial versus proself fashion across items.

### 4.2.2. Attachment

Attachment was assessed with one or two measures per assessment wave. The switch of these instruments occurred based on considerations unrelated to the current study and were thus not informed by any preliminary analysis (which we did not conduct). In all studies, attachment style was assessed in a dimensional fashion. In 2011 and 2014, we used the Adult Attachment Scale (AAS; [22]), which is a questionnaire based on Bartholomew & Horowitz [7] consisting of subscales for attachment avoidance, attachment anxiety and attachment security. In 2012, 2013 and 2014, attachment style was assessed through the Revised Experiences in Close Relationships Scale (ECR-R; [8,11]), which consists of subscales for attachment avoidance and attachment anxiety (but not for attachment security, based on the contemporary understanding that security is a derivate of avoidance and anxiety). In table 2, we report the psychometric properties of all attachment measures. Note that, in 2014, two instruments were included to measure avoidance and anxiety, which correlated 0.59 and 0.71 across instruments, respectively. Consistent with the modern multidimensional understanding of attachment security, the AAS security scale correlated $r = -0.33$, $p < 0.001$ with anxiety, and $r = -0.50$, $p < 0.001$ with avoidance across 2011 and 2014, respectively (when attachment security was measured with a separate scale). The lack of unidimensionality is also apparent from the very low Omega reliability for the security scale in 2011 and 2014, see table 2.

Importantly, our attachment measures differed from Van Lange *et al.* [1] in two ways: (i) we did not use his original instrument because better instruments have become available and (ii) we did not focus on security in the 2012 and 2013 samples for lack of a corresponding measure. Therefore, only results based on the 2011 and 2014 samples could be seen as direct replications (of the finding regarding attachment security). Having said that, we regard the ECR-R as a measure that is superior to three-dimensional instruments because it is more consistent with contemporary dimensional models of attachment.

### 4.2.3. Additional measures and positive controls

As noted, the data was collected in test-weeks. A full overview of relevant measures is reported in tables 3 and 4. We tested these measures as positive controls (by examining their reliability) *and* to investigate whether our central measures (SVO and attachment) behaved like they are expected to behave on the basis of the existing literature.

**Table 2.** Measure, year and reliabilities of the attachment measures. Note: To compute Omega Hierarchical, we used the corresponding function from the psych package [23]. In most cases, the default number of 3 group factors was retained. In some cases (e.g. the 2012 ECR avoidance scale; see syntax files uploaded on OSF), there were problems with the factor estimation (e.g. Heywood cases). In these cases, we set the number of group factors to 4, which solved these problems.

| instrument/year/scale/reference | items | reliability measure | |
| --- | --- | --- | --- |
| | | Alpha | Omega Hierarchical |
| Adult Attachment Scale (2011)—Collins & Read [22] | | | |
| avoidance | 6 | 0.71 | 0.56 |
| security | 6 | 0.47 | 0.19 |
| anxiety | 6 | 0.76 | 0.55 |
| Experiences in Close Relationships (2012)—Fraley *et al.* [24] | | | |
| avoidance | 18 | 0.90 | 0.66 |
| anxiety | 18 | 0.90 | 0.59 |
| Experiences in Close Relationships (2013)—Fraley *et al.* [24] | | | |
| avoidance | 18 | 0.90 | 0.68 |
| anxiety | 18 | 0.90 | 0.72 |
| Experiences in Close Relationships (2014)—Fraley *et al.* [24] | | | |
| avoidance | 18 | 0.91 | 0.75 |
| anxiety | 18 | 0.90 | 0.59 |
| Adult Attachment Scale (2014)—Collins & Read [22] | | | |
| avoidance | 6 | 0.65 | 0.49 |
| security | 6 | 0.36 | 0.18 |
| anxiety | 6 | 0.74 | 0.49 |

### 4.2.4. Tests of normality

We also tested the SVO-attachment data for normality via the Shapiro–Wilk test. Although this test was significant for avoidance (proself $SW = 0.98$, $p < 0.01$; prosocial $SW = 0.98$, $p < 0.01$), once significant for anxiety (proself $SW = 0.96$, $p = 0.059$; prosocial $SW = 0.99$, $p < 0.01$), significant for security (proself $SW = 0.96$, $p < 0.01$; prosocial $SW = 0.99$, $p = 0.03$), the Central Limit Theorem prescribes that distributions of means approach normality as long as the sample is sufficiently large [31]. When we plotted our distributions, the distribution indeed appeared normal for avoidance and anxiety. The distribution seemed less normal for security, which is in line with its problematic measurement quality (see our OSF file page for the folder with plots: https://osf.io/6kqzy/files/) so we proceeded with parametric testing, as was also done in the Van Lange *et al.* [1] paper.

## 5. Confirmatory results: are prosocials more securely attached?

The following results were found with the merged datasets. Given that not all questionnaires were administered in each year, sample size varied per study. Furthermore, degrees of freedom varied per analysis. We will not list these descriptors for each analysis, but instead point the reader to the data and analysis scripts that are available online under the files at our Open Science Framework page (https://osf.io/6kqzy/).

Our sample included 499 prosocials (65%) and 269 proselfs (35%) and 95 individuals who could not be classified. If participants answered more than six dilemmas in a prosocial manner, they were classified as being prosocial. Note that the number of prosocials are somewhat higher than reported in the Van Lange paper, where between 43% and 49% of the sample was classified as prosocial, $\chi^2 = 63.75$, d.f. = 1, $p < 0.001$. Note, however, that Van Lange also reported a larger number of individuals who could not be classified.

Given that attachment measures were measured at different scales (from 1 to 5 for the AAS and 1 to 7 for the ECR-R) we standardized them within each year for our aggregate (cross-sample) analysis. In our

**Table 3.** Measure, year, reliabilities and references of additional measures.

| instrument/reference/year | items | reliability measure | |
| | | Alpha | Omega Hierarchical |
|---|---|---|---|
| Self-Report Psychopathy Scale—Paulhus *et al.* (2006) [25] | | | |
| 2011 | 64 | 0.92 | 0.60 |
| 2012 | 64 | 0.92 | 0.54 |
| 2013 | 64 | 0.91 | 0.52 |
| Interpersonal Reactivity Index—Perspective Taking—Davis (1980) [26] | | | |
| 2011 | 7 | 0.78 | 0.64 |
| 2012 | 7 | 0.78 | 0.59 |
| Interpersonal Reactivity Index—Empathic Concern—Davis (1980) [26] | | | |
| 2011 | 7 | 0.82 | 0.76 |
| 2012 | 7 | 0.67 | 0.55 |
| TIPI—Openness to Experience—Gosling *et al.* (2003) [27] | | | |
| 2011 | 2 | 0.33 | n.a. |
| 2012 | 2 | 0.39 | n.a. |
| 2013 | 2 | 0.36 | n.a. |
| TIPI—Conscientiousness—Gosling *et al.* (2003) [27] | | | |
| 2011 | 2 | 0.43 | n.a. |
| 2012 | 2 | 0.37 | n.a. |
| 2013 | 2 | 0.38 | n.a. |
| TIPI—Extraversion—Gosling *et al.* (2003) [27] | | | |
| 2011 | 2 | 0.76 | n.a. |
| 2012 | 2 | 0.55 | n.a. |
| 2013 | 2 | 0.65 | n.a. |
| TIPI—Agreeableness—Gosling *et al.* (2003) [27] | | | |
| 2011 | 2 | 0.20 | n.a. |
| 2012 | 2 | −0.06 | n.a. |
| 2013 | 2 | 0.06 | n.a. |
| IPI—Neuroticism—Gosling *et al.* (2003) [27] | | | |
| 2011 | 2 | 0.63 | n.a. |
| 2012 | 2 | 0.67 | n.a. |
| 2013 | 2 | 0.60 | n.a. |
| Self-Control—Finkenauer *et al.* (2005) [28] | | | |
| 2012 | 11 | 0.74 | 0.40 |
| 2013 | 11 | 0.71 | 0.51 |

2014 sample, we furthermore aggregate across the avoidance and anxiety dimensions of both available measures. We also analyse all scales separately and report the results in footnotes.[1] For sake of consistency, we report ANOVAs in all our replication analyses.

---

[1]In addition, in order to discover whether the type of questionnaire mattered in its relation to SVO, we also added a factor 'assessment year' in our dataset. The ECR and AAS were both included in 2014. We did not detect significant interaction effects between social value orientation and test-week year on attachment avoidance ($F_{1,761} = 0.75$, $p = 0.39$), attachment anxiety ($F_{1,762} = 2.15$, $p = 0.15$), with the average of AAS and ECR included in 2014. The same non-significant interaction was found for attachment security ($F_{1,410} = 1.56$, $p = 0.21$).

**Table 4.** Measure, year, reliabilities and references of additional measures. Note: We only included scales in this table (and our auxiliary analyses reported on the Open Science Framework) that were measured in at least 2 years, for reasons of power. To compute Omega Hierarchical, we used the corresponding function from the psych package [23]. In most cases, the default number of 3 group factors was retained. In some cases (e.g. the empathic concern scale; see syntax files uploaded on OSF), there were problems with the factor estimation (e.g. Heywood cases). In these cases, we set the number of group factors to 4, which solved factor estimation problems. For all subscales that only contained two items (i.e. the TIPI subscales) Omega cannot be computed.

| instrument/reference/year | items | reliability measure | |
|---|---|---|---|
| | | Alpha | Omega Hierarchical |
| Behavioral Inhibition System—Franken *et al.* (2005) [29] | | | |
| 2013 | 7 | 0.79 | 0.66 |
| 2014 | 7 | 0.85 | 0.75 |
| Behavioral Approach System—Franken *et al.* (2005) [29] | | | |
| 2013 | 7 | 0.73 | 0.38 |
| 2014 | 7 | 0.74 | 0.59 |
| Rosenberg Self Esteem—Rosenberg (1965) [30] | | | |
| 2011 | 10 | 0.91 | 0.82 |
| 2013 | 10 | 0.89 | 0.73 |
| 2014 | 10 | 0.87 | 0.75 |

In the Van Lange *et al.* [1] paper, results were analysed by contrasting prosocials versus proselfs (individualists and competitors, jointly). The authors reported that people who are more prosocial (as compared to 'proselfs') were more secure in their attachment towards a variety of close others.[2] We did not find higher levels of attachment security for prosocials ($M = 0.05$, s.d. $= 1.00$) than proselfs ($M = -0.10$, s.d. $= 1.07$) in the combined 2011 or 2014 samples; however, Cohen's $d = 0.15$, ($F_{1,412} = 2.02$, $p = 0.16$).[3] Van Lange *et al.* [1] also reported differences on attachment anxiety and avoidance, which is more in line with contemporary dimension models of attachment styles. For the four merged samples we could also not support the hypothesis that prosocial ($M = 0.03$, s.d. $= 1.01$) people are less anxious than proselfs ($M = 0.01$, s.d. $= 1.01$), Cohen's $d = 0.03$, $F_{1,764} = 0.11$, $p = 0.74$, or that prosocials ($M = -0.02$, s.d. $= 1.01$) are less avoidant in their attachment than proselfs ($M = 0.05$, s.d. $= 1.03$), Cohen's $d = 0.07$, $F_{1,763} = 0.86$, $p = 0.36$. When we tested for the differences between prosocial ($M = -0.01$, s.d. $= 0.85$) and proself ($M = -0.03$, s.d. $= 0.87$) with the ECR reverse-scored and averaged as proxy for security, we also found no difference, Cohen's $d = 0.03$, $F_{1,763} = 0.12$, $p = 0.73$. We also considered testing the relationship between SVO using a continuous measure instead of a categorical measure. However, because the data were binomially distributed, we could not support analysing SVO as continuous.

## 6. Discussion

In a large sample study from four years ($N = 879$) we found no relationship between social value orientation and attachment, and we could thus not replicate Van Lange *et al.* [1], who found that people who are more prosocial (versus proself) are more secure (or, alternatively, less avoidant and less anxious) in their attachment. Thus, with a total $N = 879$ participants, we could not confirm the hypothesis that SVO is related to attachment security.

There are three possible explanations for the discrepancy. First, it is possible that our study was not well conducted, including the fact that our attachment measures differed. This is of course possible, but

---

[2]Van Lange *et al.* [1] also report the analysis with participants' biological sex as covariate, but did not find any differences. We nevertheless ran this test again and find again no differences for SVO on avoidance with sex as covariate ($F_{1,739} = 0.94$, $p = 0.33$), nor we do find any differences for SVO on anxiety with sex as covariate ($F_{1,740} = 0.01$, $p = 0.75$). The same was true for SVO on security ($F_{1,409} = 2.03$, $p = 0.16$).

[3]When we analysed attachment security by year, we found no significant effects for 2011 ($F_{1,236} = 3.58$, $p = 0.06$) or for 2014 ($F_{1,174} < 0.01$, $p = 0.98$).

we could not identify any psychometric indicators to support such a claim. Our measures were more reliable than Van Lange et al.'s [1] measures, which were based on the very early stages of psychometric research of attachment measures [32]. Moreover, the dimensionality of our measures partly differed: whereas Van Lange et al. [1] measured three dimensions (including security), contemporary dimensional approaches do not regard security as a separate dimension but instead consisting of combined low levels of avoidance and anxiety (which was confirmed by our correlational analysis). It has since also become more widely known that scale construction for attachment security is therefore problematic from this perspective, probably resulting in the low alphas that were reported in Van Lange et al. Because security is defined by two dimensions (low anxiety and low avoidance), the alpha of a security composite will necessarily be modest (Fraley 2017, personal communication). From a measurement perspective, it is problematic that Van Lange et al. [1] only found significant results for the unreliable security scale and only marginally significant results for the more reliable avoidance and anxiety scales.

One could further discuss what constitutes a solid replication (e.g. [33], for a discussion). In our view, in certain instances (e.g. when measurement quality has improved) a solid (conceptual) replication consists of successfully testing the idea behind the study. Science is a cumulative enterprise, and measures naturally improve over time. What should replicators do in case original findings are based on outdated instruments (needless to say, they might not have been outdated at the time of the original study)? We decided to measure the construct of attachment according to state-of-the-art methods and with translations that were better done than the original, even if it meant to give up on the security dimension in favour of its two-constituting avoidance and anxiety dimensions. Nevertheless, we *still* tested the relationship with more outdated scales and also found nothing.

Do note, however, that in 2011, a psychometrically superior measure—the slider measure—for SVO became available [34]. Our data collection ran from 2011–2014, and the old measure, with nominal categories, was still included in our test batteries (despite the superior measure having been available) and this was a shortcoming of our research. Because the slider measure has been shown to be superior to our measure (e.g. [35]), whether or not the slider measure does correlate with attachment is a question for future research.

We think the soundness of our approach is evident from the fact that our attachment measures did not only have superior reliability as compared to the original study but also demonstrated construct validity in conducted auxiliary analyses (reported on the Open Science Framework: https://osf.io/6n3ta/). These analyses clearly showed robust patterns (achieved through split-half cross validation) of attachment dimensions correlating in expected ways with theoretically related constructs. Reporting being less avoidant in one's attachment related to scoring higher on empathic concern, perspective taking, self-esteem, agreeableness and extraversion, and scoring lower on psychopathic tendencies. Reporting being less anxious in one's attachment related to scoring lower on behavioural inhibition and neuroticism, and higher on perspective-taking, self-esteem, openness to experience, extraversion and self-control. Importantly, we did not detect interaction effects between measures and assessment year. Note also that when we did use a measure that included a scale of attachment security, we also did not replicate the original Van Lange et al. [1] results. Still, it is possible that future measurement developments will contribute to psychometrically superior measures of attachment security that might provide additional replication evidence regarding its link with SVO.

Second, there might be changes in sample composition from the original to the replication study. Besides some small changes (i.e. our sample was from a smaller-sized provincial town instead of Amsterdam, a large and international metropolis), we did find that our sample was slightly more proself than Van Lange. It could be speculated that the urban versus provincial status of the samples might have contributed to this difference, but we should also note that Tilburg is itself a medium-sized (for Dutch standards) industrial town that is only a 1 h drive from Amsterdam. Variance restriction might have been an issue, although we do not think this accounts for the lack of replication. Finally, we note that our sample size was considerably larger than the original and our sensitivity analyses showed that we were well positioned to detect a small effect.

A final possibility is that the lack of replication of the original study was made possible by problems with the measurement of attachment. First and foremost, the reliabilities of Van Lange et al.'s [1] Study 1's only attachment measure providing a significant result was low, introducing abundant measurement error that might have contributed to a lack of robustness (depending on which items were included, the alpha ranged between 0.46 and 0.57). The other scales in their Study 1 (anxiety and avoidance, alphas between 0.66 and 0.68) produced only marginal differences ($p = 0.09$ and $p = 0.06$, respectively) in a sample of 573 participants. These marginal differences were obtained while

controlling for gender *and* after dropping items from the attachment scale related to feelings about one's partner (yet in Study 2 the effect was present for partner-specific attachment). In terms of validity, there might also have been problems with the attachment measure's translation. In Van Lange *et al.*'s [1] paper, the authors reported that they asked participants about such items as 'I find it easy to trust others,' 'I find it easy to get close to others', 'I feel comfortable having other people depend on me', 'I am nervous when anyone gets too close' and 'I find that other people don't want to get as close as I would like'. We did not find these items to be fully representative of their actual measurement, which we obtained from the first author of the Van Lange *et al.* [1] paper (see our appendix). Attachment items related to reliance on others (a central characteristic of being attached) were dropped, and the items that were retained seemed to rely on a reciprocal exchange of feelings (e.g. 'I do not mind at all to become very personal (e.g. to exchange feelings) with others'; see for the full list appendix A), rather than a reliance or dependence on others that stresses the 'merging of self and other'.

While preparing this report, we also learnt that Klein *et al.* [16] studied another effect of the Van Lange paper (Study 3; the relationship between having siblings and being prosocial or not) and this effect was also not replicated. *Post hoc*, after knowing the outcome of these replication studies, one could argue that Studies 1–3 are not vital to the 1997 paper (e.g. [36]), with which we strongly disagree. These studies provided a proof-of-concept to the SVO measure, as has also become clear from subsequent studies citing Van Lange *et al.* [1]. Rusbult & Van Lange [37] for example indicate that 'The adult attachment literature suggests that the intrapersonal and interpersonal adaptations acquired in childhood are carried into adult interactions' (p. 364), while another highly cited article stated that 'SVO might be related to perceived closeness because of its relationship with attachment styles' [38, p. 1082]. Collins & Feeney [39] similarly stated that 'adults who are higher in attachment-related anxiety…are less likely to hold a prosocial interpersonal orientation' (p. 1068), while De Dreu and colleagues [40] concluded that 'Social motives may be rooted, for example, in individual differences in social value orientations' (p. 890). In other words, Van Lange *et al.*'s [1] first two studies were crucial for how social dilemma theorists think about SVO, and the Van Lange *et al.* [1] article provided an important proof-of-concept for the SVO measure.

Where do we go from here? Going into this project, we did not think it was an unreasonable assumption that attachment was in some way related to SVO. After all, both relate to the honesty/humility factor of the HEXACO [41] and the same trait is inversely related to anxious attachment [42]. Of course, it might be that other replicators still confirm the link between SVO and secure attachment. We think, however, that alternative links are theoretically more likely. Some of our auxiliary analyses suggested that being more prosocial related to all kinds of seemingly positive qualities, like higher scores on agreeableness, empathic concern and perspective-taking, and lower scores on psychopathic tendencies. These could point, instead of a relation to attachment security, to a self-regulation style known in the literature as 'unmitigated communion' [43]. It is a type of communion often confused with an orientation on being affectionate and attuned to others' need, but, unlike real communion, unmitigated communion is a 'focus on others to the detriment of the self' [43, p. 121]. Future research is required to shed light on whether being prosocial on the SVO measure is the kind of unmitigated communion we propose.

What are then the developmental origins of SVO? Although we think this is an exciting question, we simply don't know. In any case, given that this paper was the bedrock of an entire literature and that Van Lange *et al.*'s [1] Studies 1–3 are now on a shaky foundation, we think that these developmental origins as well as the measure of SVO deserve a higher level of evidence than is currently available, in order to rebut the worries that have arisen about the replicability of findings in psychology more broadly. We continue to think that SVO is a very interesting construct that will benefit from a broadened basis of empirical correlates and therefore call for a renewed look at the developmental origins of SVO. More generally, we think that the present replication effort also illustrates the importance of investing in consensual and psychometrically sound instruments to measure individual differences like attachment and SVO. We think one underappreciated but important factor in efforts to increase the replicability of psychological research is the improvement of measures to assess fundamental constructs.

Ethics. We analysed secondary data and thus did not do the ethics part ourselves. Nevertheless, ethical approval was not obtained for the study; ethical approval for behavioural science was not commonplace in The Netherlands at the time the data were collected.

Data accessibility. Data, analysis syntax, and auxiliary analyses available on the article's project page: https://osf.io/6kqzy/.

Authors' contribution. H.IJ. retrieved the data files from the test-weeks at Tilburg University. H.IJ. then conducted the first analyses and wrote the first draft of the manuscript. J.J.A.D. then wrote a new version of the analyses code and

provided several reviews of the manuscript. H.IJ. then rewrote the analyses code again and rewrote the manuscript. H.IJ. and J.J.A.D. then went back and forth until the final product was submitted. H.IJ. coordinated submission of the manuscript to the journal and registered the analyses (which had already been conducted) after Stage 1 approval.
Competing interests. We have no competing interests.
Funding. This research was supported by a Veni grant of the Netherlands Organization for Scientific Research (NWO) (016.145.049) and a French National Research Agency 'Investissements d'avenir' programme grant no. (ANR-15-IDEX-02) both awarded to H.IJ.
Acknowledgements. This article received results-blind *In Principle Acceptance* (IPA) at Royal Society Open Science. Following IPA, H.IJ. pre-registered the IPA at the OSF (https://osf.io/wxgv8/). Please note that this pre-registration was done after data analysis.

# Appendix A

Van Lange *et al.*'s [1] Study 1 original Dutch attachment questionnaire, with our translation included and the original English items (supplied by Van Lange).

| Van Lange *et al.*'s [1] items | our back-translation | original items in English, supplied by Van Lange *et al.* [1] |
|---|---|---|
| Ik vertrouw anderen vrij snel. | I trust others relatively quickly. | I find it easy to trust others. |
| Ik heb er helemaal geen moeite mee om erg persoonlijk (b.v., uitwisselen van gevoelens) met anderen te worden. | I do not mind at all to become very personal (e.g. to exchange feelings) with others. | I find it easy to get close to others. |
| Ik voel me op mijn gemak met anderen, ook al weet ik dat ik deels afhankelijk ben van deze anderen. | I feel at ease with others, even though I know that I am partly dependent of these others. | I feel comfortable depending on other people.[a] |
| Ik heb er helemaal geen moeite mee als anderen van mij afhankelijk zijn. | I do not mind at all if others are dependent on me. | I feel comfortable having other people depend on me.[a] |
| Als anderen dat graag willen, sta ik best open voor een meer persoonlijke relatie. | If others would like to, I am open to have a more personal relationship. | I don't often worry about someone getting too close to me. |
| Anderen zijn vaak minder persoonlijk dan ik zou willen. | Others are often less personal than I would like. | I find that other people don't want to get as close as I would like. |
| Ik ben wel eens bang dat sommige van mijn vrienden mij in steek laten. | I sometimes am afraid that some of my friends desert me. | I worry that a love partner might not really love me.[a] |
| Vooral wanneer het er echt op aankomt, ben ik bang dat sommige vrienden me in de steek laten. | When it really comes down to it, I am afraid that some friends desert me. | I don't often worry about being abandoned. |
| Ik wil graag een dieper (b.v., meer uitwisseling van gevoelens) contact met anderen, maar anderen staan daar niet altijd voor open. | I desire a deeper (for example, an exchange of feelings) contact with others, but others are not always open for that. | I am uncomfortable being close to others. |

(*Continued.*)

| Van Lange *et al.*'s [1] items | our back-translation | original items in English, supplied by Van Lange *et al.* [1] |
|---|---|---|
| Ik vind het onprettig om persoonlijk te zijn met anderen. | I find it unpleasant to be personal with others. | I am nervous when anyone gets too close. |
| Ik voel me vaak niet op mijn gemak als anderen te persoonlijk worden. | I often don't feel at ease when other people become too personal. | I worry that love partners might want me to be more intimate than I feel comfortable being.[a] |
| Ik ben wel eens bang dat anderen intiemer en persoonlijker willen zijn dan dat ik zelf prettig vind. | I am sometimes afraid that others become more intimate and personal than I would find pleasant. | I find it difficult to depend on others. |
| Ik pas goed op dat ik niet te veel persoonlijke zaken aan anderen vertel. | I monitor well that I don't tell too many personal details to others. | I want to merge completely with another person. |

[a]Dropped in Van Lange due to low reliability. For two of the items, Van Lange indicated that he found 'afhankelijk' (=dependent) too strong in the Dutch version and therefore dropped the items.

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
