## [Reviewer comments · Royal Society Open Science]

Review History

RSOS-181575.R0 (Original submission)

Review form: Reviewer 1

Do you have any ethical concerns with this paper?

No

Have you any concerns about statistical analyses in this paper?

Yes

Recommendation?

Major revision

Comments to the Author(s)

Please see the attached file (Appendix A).

Review form: Reviewer 2

Do you have any ethical concerns with this paper?

No

Have you any concerns about statistical analyses in this paper?

No

Recommendation?

Accept in principle

Comments to the Author(s)

This is a sound replication of previous research by Van Lange et al. (1997) on social value orientation and adult attachment. I have two minor suggestions that I think would help improve the paper. First, the authors seem to be implying that the large number of citations that the original article received is a marker of the paper's influence. But my sense, from examining some of the articles that have cited the original is that those citations are not for the attachment-SVO findings/claims, but rather simply a citation for the SVO idea itself and some of the methods that were used to study it. I think it is potentially misleading to imply that the attachment-SVO findings have been influential in the field.

Second, on page 5, I would caution against suggesting that adult attachment patterns are a function of "early" experience. The statement implicitly suggests that later experiences are not relevant. I doubt the authors believe that, but that is one way to interpret the use of the term "early" rather than, say, "prior" or "a history of interpersonal experiences."

I think the methods and analytic strategies are sound. I'm glad the sample size is relatively large.

Decision letter (RSOS-181575.R0)

08-Nov-2018

Dear Dr IJzerman,

The Editors assigned to your Stage 1 Replication submission ("Social Value Orientation and Attachment: A Replication and Extension of Van Lange et al. (1997)") have now received comments from reviewers. We would like you to revise your paper in accordance with the referee and editors suggestions which can be found below (not including confidential reports to the Editor).

Please submit a copy of your revised paper within three weeks (i.e. by the The author due date is unavailable). If deemed necessary by the Editors, your manuscript will be sent back to one or more of the original reviewers for assessment.

When submitting your revised manuscript, you must respond to the comments made by the referees and upload a file "Response to Referees" in the "File Upload" step. Please use this to document how you have responded to the comments, and the adjustments you have made. In order to expedite the processing of the revised manuscript, please be as specific as possible in your response.

Once again, thank you for submitting your manuscript to Royal Society Open Science and I look forward to receiving your revision. If you have any questions at all, please do not hesitate to get in touch. Full author guidelines may be found at <http://rsos.royalsocietypublishing.org/page/replication-studies#AuthorsGuidance>.

Kind regards,
Professor Chris Chambers
Royal Society Open Science
openscience@royalsociety.org

on behalf of Professor Chris Chambers (Registered Reports Editor, Royal Society Open Science)
openscience@royalsociety.org

Associate Editor Comments to Author (Professor Chris Chambers):

Two expert reviewers have now appraised the manuscript. As you will see, both are positive about the submission but note several areas requiring attention. The most substantial concern (Reviewer #1) is the need for greater detail about the analyses and certain aspects of the instruments. In addition to this major issue, both reviewers offer a range of constructive suggestions for clarifying and justifying specific arguments.

Comments to Author:

Reviewer: 1

Comments to the Author(s)

Please see the attached file.

Reviewer: 2

Comments to the Author(s)

This is a sound replication of previous research by Van Lange et al. (1997) on social value orientation and adult attachment. I have two minor suggestions that I think would help improve the paper. First, the authors seem to be implying that the large number of citations that the original article received is a marker of the paper's influence. But my sense, from examining some of the articles that have cited the original is that those citations are not for the attachment-SVO findings/claims, but rather simply a citation for the SVO idea itself and some of the methods that were used to study it. I think it is potentially misleading to imply that the attachment-SVO findings have been influential in the field.

Second, on page 5, I would caution against suggesting that adult attachment patterns are a function of "early" experience. The statement implicitly suggests that later experiences are not relevant. I doubt the authors believe that, but that is one way to interpret the use of the term "early" rather than, say, "prior" or "a history of interpersonal experiences."

I think the methods and analytic strategies are sound. I'm glad the sample size is relatively large.

Author's Response to Decision Letter for (RSOS-181575.R0)

See Appendix B.

RSOS-181575.R1 (Revision)

Review form: Reviewer 1

Do you have any ethical concerns with this paper?

No

Have you any concerns about statistical analyses in this paper?

No

Recommendation?

Accept in principle

Comments to the Author(s)

The authors have adequately addressed all of my comments from the first review.

Review form: Reviewer 2

Do you have any ethical concerns with this paper?

No

Have you any concerns about statistical analyses in this paper?

No

Recommendation?

Accept in principle

Comments to the Author(s)

This looks solid. I made a few suggestions in my last review and I appreciate the authors taking them into consideration.

Decision letter (RSOS-181575.R1)

05-Dec-2018

Dear Dr IJzerman

On behalf of the Editor, I am pleased to inform you that your Stage 1 Replication submission RSOS-181575.R1 entitled "Social Value Orientation and Attachment: A Replication and Extension

of Van Lange et al. (1997)" has been accepted in principle for publication in Royal Society Open Science. The reviewers' and editors' comments are included at the end of this email.

You may now progress to Stage 2 and complete the study as approved.

You must now register your approved protocol on the Open Science Framework (<https://osf.io/rr>), either publicly or privately under embargo until submission of the Stage 2 manuscript. Please note that a time-stamped, independent registration of the protocol is mandatory under journal policy, and manuscripts that do not conform to this requirement cannot be considered at Stage 2. The protocol should be registered unchanged from its current approved state. Please include a URL to the protocol in your Stage 2 manuscript.

Following completion of your study, we invite you to resubmit your paper for peer review as a Stage 2 Replication. Please note that your manuscript can still be rejected for publication at Stage 2 if the Editors consider any of the following conditions to be met:

- The Introduction and methods deviated from the approved Stage 1 submission (required).
- The authors' conclusions were not considered justified given the data.

We encourage you to read the complete guidelines for authors concerning Stage 2 submissions at: <http://rsos.royalsocietypublishing.org/page/replication-studies#AuthorsGuidance>. Please especially note the requirements for data sharing and that withdrawing your manuscript will result in publication of a Withdrawn Registration.

Once again, thank you for submitting your manuscript to Royal Society Open Science and I look forward to receiving your Stage 2 submission. If you have any questions at all, please do not hesitate to get in touch. We look forward to hearing from you shortly with the anticipated submission date for your stage two manuscript.

Kind regards,
Andrew Dunn
Royal Society Open Science
openscience@royalsociety.org

on behalf of Professor Chris Chambers (Registered Reports Editor, Royal Society Open Science)
openscience@royalsociety.org

Editor Comments to Author (Professor Chris Chambers):

Thanks you for your revised submission. Both Stage 1 reviewers are satisfied with the revision and it can now be awarded in principle acceptance and progress to Stage 2.

Reviewers' comments to Author:

Reviewer: 2

Comments to the Author(s)

This looks solid. I made a few suggestions in my last review and I appreciate the authors taking them into consideration.

Reviewer: 1

Comments to the Author(s)

The authors have adequately addressed all of my comments from the first review.

Author's Response to Decision Letter for (RSOS-181575.R1)

See Appendix C.

RSOS-181575.R2 (Revision)

Review form: Reviewer 1

Do you have any ethical concerns with this paper?

No

Have you any concerns about statistical analyses in this paper?

No

Recommendation?

Accept with minor revision

Comments to the Author(s)

I have one minor and one major point that I would like the authors to address before I can recommend the manuscript for publication.

Minor

The authors should explicate the number of individuals that could not be classified. Also there is currently no description of the criterion/cutoff used for the classification.

Major

Following my request from the first stage review, the authors have added a footnote on the availability of better SVO measures. However, I do not think that this should be put in a footnote, it deserves to be part of the main body of the discussion. First, the very philosophy of the authors (which I endorse!) is that replication should make use of the development of better measures. To be consistent with this thinking, the authors should not only stress to what extent their paper meets this standard but also point to their failures of meeting it (although it was clearly not a fault as new SVO instruments were freshly developed when data collection commenced). This does not weaken the paper but instead makes it more consistent and convincing. Second, the authors' argument that their SVO scale was highly consistent based on KR20 is at least difficult. On the one hand, this statistic rests on an artificial dichotomization of participants' answers. On the other hand, it is not surprising to get a high KR20s given that the triple dominance scales essentially confront participants with three highly similar choices (Murphy & Ackermann, 2014, PERS SOC PSYCHOL REV). Third (and related), the very point about newly developed SVO scales (e.g., Murphy, Ackermann, & Handgraaf, 2011, JUDGM DECIS MAK) is the insight that gradual differences in SVO can matter much more than nominal classifications - classifications can even conceal meaningful differences due to SVO (for illustrations of this important point, see Bieleke, Gollwitzer, Oettingen, & Fischbacher, 2017, J BEHAV DECIS MAKING; Fiedler, Glöckner, Nicklisch, & Dickert, 2013, ORGAN BEHAV HUM DEC). To me, this insight seems at least as important as the new developments regarding attachment scales stressed in the present paper. Addressing this point more thoroughly would make the paper a much more useful contribution for researchers who care about SVO. Moreover, it provides a promising avenue for

future research - which, for instance, might examine whether gradual SVO differences relate more robustly to attachment.

Review form: Reviewer 2

Do you have any ethical concerns with this paper?

No

Have you any concerns about statistical analyses in this paper?

No

Recommendation?

Accept as is

Comments to the Author(s)

I believe the authors have done an excellent job at analyzing the data and providing a fair and rigorous test of the key idea (i.e., that there is an association between attachment security and SVO).

I also believe the authors have summarized their findings well, discussed them in valuable ways, and have been appropriately cautionary about what has and has not been learned from this research.

In short, I think this manuscript is ready to be published.

Decision letter (RSOS-181575.R2)

07-Feb-2019

Dear Dr IJzerman

On behalf of the Editor, I am pleased to inform you that your Stage 2 Replication submission RSOS-181575.R2 entitled "Social Value Orientation and Attachment: A Replication and Extension of Van Lange et al. (1997)" has been accepted for publication in Royal Society Open Science subject to minor revision in accordance with the referee suggestions. Please find the referees' comments at the end of this email.

The reviewers and Subject Editor have recommended publication, but also suggest some minor revisions to your manuscript. Therefore, I invite you to respond to the comments and revise your manuscript.

Please also ensure that all the below editorial sections are included where appropriate (a non-exhaustive example is included in an attachment):

- Ethics statement

If your study uses humans or animals please include details of the ethical approval received, including the name of the committee that granted approval. For human studies please also detail

whether informed consent was obtained. For field studies on animals please include details of all permissions, licences and/or approvals granted to carry out the fieldwork.

- Data accessibility

If you wish to submit your supporting data or code to Dryad (<http://datadryad.org/>), or modify your current submission to dryad, please use the following link:
<http://datadryad.org/submit?journalID=RSOS&manu=RSOS-181575.R2>

- Competing interests

- Authors' contributions

- Acknowledgements

- Funding statement

Because the schedule for publication is very tight, it is a condition of publication that you submit the revised version of your manuscript within 7 days (i.e. by the 15-Feb-2019). If you do not think you will be able to meet this date please let me know immediately.

To revise your manuscript, log into <https://mc.manuscriptcentral.com/rsos> and enter your Author Centre, where you will find your manuscript title listed under "Manuscripts with Decisions". Under "Actions," click on "Create a Revision." You will be unable to make your

revisions on the originally submitted version of the manuscript. Instead, revise your manuscript and upload a new version through your Author Centre.

- 1) A text file of the manuscript (tex, txt, rtf, docx or doc), references, tables (including captions) and figure captions. Do not upload a PDF as your "Main Document".
- 2) A separate electronic file of each figure (EPS or print-quality PDF preferred (either format should be produced directly from original creation package), or original software format)
- 3) Included a 100 word media summary of your paper when requested at submission. Please ensure you have entered correct contact details (email, institution and telephone) in your user account
- 4) Included the raw data to support the claims made in your paper. You can either include your data as electronic supplementary material or upload to a repository and include the relevant DOI within your manuscript
- 5) Included your supplementary files in a format you are happy with (no line numbers, Vancouver referencing, track changes removed etc) as these files will NOT be edited in production

Kind regards,
Professor Chris Chambers
Royal Society Open Science
openscience@royalsociety.org

Associate Editor Comments to Author (Professor Chris Chambers):

Associate Editor: 1

Comments to the Author:

The manuscript was returned to the two expert reviewers who assessed it at Stage 1. Both are positive about the submission, with Reviewer 1 recommending some elaboration of the Discussion, and Reviewer 2 recommending immediate acceptance. Please attend carefully to the comments of Reviewer 1. Provided these concerns are fully addressed in revision, full acceptance should be forthcoming without requiring further in-depth review.

Reviewers' comments to Author:

Reviewer: 1

Comments to the Author(s)

I have one minor and one major point that I would like the authors to address before I can recommend the manuscript for publication.

Minor

The authors should explicate the number of individuals that could not be classified. Also there is currently no description of the criterion/cutoff used for the classification.

Major

Following my request from the first stage review, the authors have added a footnote on the availability of better SVO measures. However, I do not think that this should be put in a footnote, it deserves to be part of the main body of the discussion. First, the very philosophy of the authors (which I endorse!) is that replication should make use of the development of better measures. To be consistent with this thinking, the authors should not only stress to what extent their paper meets this standard but also point to their failures of meeting it (although it was clearly not a fault as new SVO instruments were freshly developed when data collection commenced). This does not weaken the paper but instead makes it more consistent and convincing. Second, the authors' argument that their SVO scale was highly consistent based on KR20 is at least difficult. On the one hand, this statistic rests on an artificial dichotomization of participants' answers. On the other hand, it is not surprising to get a high KR20s given that the triple dominance scales essentially confront participants with three highly similar choices (Murphy & Ackermann, 2014, *PERS SOC PSYCHOL REV*). Third (and related), the very point about newly developed SVO scales (e.g., Murphy, Ackermann, & Handgraaf, 2011, *JUDGM DECIS MAK*) is the insight that gradual differences in SVO can matter much more than nominal classifications - classifications can even conceal meaningful differences due to SVO (for illustrations of this important point, see Bieleke, Gollwitzer, Oettingen, & Fischbacher, 2017, *J BEHAV DECIS MAKING*; Fiedler, Glöckner, Nicklisch, & Dickert, 2013, *ORGAN BEHAV HUM DEC*). To me, this insight seems at least as important as the new developments regarding attachment scales stressed in the present paper. Addressing this point more thoroughly would make the paper a much more useful contribution for researchers who care about SVO. Moreover, it provides a promising avenue for future research - which, for instance, might examine whether gradual SVO differences relate more robustly to attachment.

Reviewer: 2

Comments to the Author(s)

I believe the authors have done an excellent job at analyzing the data and providing a fair and rigorous test of the key idea (i.e., that there is an association between attachment security and SVO).

I also believe the authors have summarized their findings well, discussed them in valuable ways, and have been appropriately cautionary about what has and has not been learned from this research.

In short, I think this manuscript is ready to be published.

Author's Response to Decision Letter for (RSOS-181575.R2)

See Appendix D.

Decision letter (RSOS-181575.R3)

20-Feb-2019

Dear Dr IJzerman:

It is a pleasure to accept your Stage 2 Replication entitled "Social Value Orientation and Attachment: A Replication and Extension of Van Lange et al. (1997)" in its current form for publication in Royal Society Open Science.

Congratulations on having the (joint) first accepted Replication article at the journal.

On behalf of the Editors of Royal Society Open Science, we look forward to your continued contributions.

on behalf of Professor Chris Chambers (Subject Editor)
openscience@royalsociety.org

Appendix A

Sunday, October 28, 2018 5:11 PM

The authors describe four waves of data collection aiming at a replication of van Lange et al.'s (1997; Studies 1 and 2) finding that social value orientation (SVO) is associated with adult attachment style. It is a Stage 1 replication study with the blinded results of an already completed data collection. I find this research timely and interesting, although I had some remarks that I would like the authors to address. Below are my thoughts and remarks regarding the different reviewer questions.

1. Stage 1 Primary Criterion #1: Whether the authors provide a sufficiently clear and detailed description of the methods for another researcher to closely replicate the proposed experimental procedures and analysis pipeline, and to prevent undisclosed flexibility in the experimental procedures or analysis pipeline.

* The authors report the measures included in all four waves of data collection. They should make explicit whether they used the exact same triple-dominance scale as van Lange et al. to measure SVO. This seems to be the case but it should be explicit and the resulting effective number of participants (including non-classifiable participants) should be mentioned.

* The authors want to make their data and analysis syntax publicly available. However, I cannot check this because the link has been redacted for review.

* Regarding the analysis pipeline, the central piece of information is missing: How will the association between SVO and attachment style be tested? Will the authors use the same approach as van Lange et al. (e.g., MANOVA, controlling for gender) or a different one? Similarly, will SVO be determined in the same way as by van Lange et al. (e.g., removing not classifiable participants)? If not, what is changed and why?

2. Stage 1 Primary Criterion #2: Whether the manuscript describes a sufficiently valid (i.e. close) and robust (e.g. statistically powerful) replication of the original study methods and rationale to provide an indication of replicability.

* The described experiment closely replicates van Lange et al., except from different measures of attachment. According to the authors, these measures are psychometrically more sound (i.e., reliable, valid) than those used by van Lange et al.

* The authors convincingly describe that van Lange et al. used a poor measure of attachment, which might have produced spurious associations with SVO. This clearly indicates the need for a replication study.

* The original study already rested on a large sample and the authors' sample even exceeds this number. The sensitivity analysis is convincing and the required analyses should be sufficiently powered.

3. Stage 1 Secondary Criterion #1: The logic, rationale, and plausibility of the proposed hypotheses.

* It is not clear whether the authors expect van Lange et al.'s study to replicate or not.

* Otherwise, the authors' assumptions seem plausible to me.

4. Stage 1 Secondary Criterion #2: The soundness of the methodology and analysis pipeline.

* I have one major issue with this research: While the authors focus on better measures of attachment, they seem to have missed crucial developments regarding the assessment of SVO (see Murphy, Ackermann, & Handgraaf, 2011; Murphy & Ackermann, 2013). The scale used by van Lange et al. has meanwhile been shown to be clearly dominated by continuous SVO scales, and the use of nominal categories is commonly discouraged for theoretical reasons (e.g., Fiedler, Glöckner, Nicklisch, & Dickert, 2013). This said, I want to stress that I like the authors' attempt to test whether an important empirical finding still holds when improved measures are used, rather than literally repeating the same experiment. However, this approach would have required to measure SVO in a modern and empirically sound way as well. This aspect reduces my enthusiasm about the described experiment. Since the experiment has already been conducted, the authors should at least address this limitation later in the discussion of their manuscript.

* To re-iterate, I feel that the analysis pipeline is not described in sufficient detail to determine whether or not it is sound.

5. Stage 1 Secondary Criterion #3: Whether the authors have considered sufficient outcome-neutral conditions (e.g. absence of floor or ceiling effects; positive controls; other quality checks) for ensuring that the results obtained are able to test the stated hypotheses.

* It would have desirable to have van Lange et al.'s original instruments in these experiments as well. This way, the authors could have tested whether differences in the findings reflect differences in the applied measures or result from other differences in the experimental protocol.

* Otherwise, the authors' approach seems sound to me.

Appendix B

Dear Dr. Chambers,

We would like to thank you for your and the reviewers' thorough attention to our work. Below we respond to the reviewers' concerns one-by-one. To facilitate review, we submit a version with track changes on and one with track changes accepted.

Reviewer 1: * The authors report the measures included in all four waves of data collection. They should make explicit whether they used the exact same triple-dominance scale as van Lange et al. to measure SVO. This seems to be the case but it should be explicit and the resulting effective number of participants (including non-classifiable participants) should be mentioned.

Authors' Reply: We thank the reviewer for pointing this out. We have changed a sentence in the procedure to clarify this point: "In each year SVO was assessed by the same nine-item triple dominance measure used in the original (Messick & McClintock, 1968; see also Van Lange et al., 1997)."

Reviewer 1: The authors want to make their data and analysis syntax publicly available. However, I cannot check this because the link has been redacted for review.

Authors' Reply: We apologize for this. Because our results were already posted with our syntax, we could not link to our page. To facilitate further review, I link to the analysis scripts in this letter from my dropbox:

Creating SVO/Attachment Scales:

<https://www.dropbox.com/s/uqqxu4q9d9c3hfh/01%20create%20SVO%20attachment%20scales.R?dl=0>

Creating Covariate Scales (other scales reported as positive controls):

<https://www.dropbox.com/s/hjwlsx66ypa8get/02%20create%20covariates%20scales.R?dl=0>

Main Analyses:

<https://www.dropbox.com/s/ctekjpnzgba77yu/03%20Analyses%20%281%29.R?dl=0>

Partial Correlations of Avoidance and SVO (reported on project page, not in manuscript):

<https://www.dropbox.com/s/v7b6rjyddj7c2zw/04%20Partial%20Correlations%20Avoidance%20%281%29.R?dl=0>

Partial Correlations of Anxiety and SVO (reported on project page, not in manuscript):

<https://www.dropbox.com/s/ep4v9498gliv1ft/05%20Partial%20Correlations%20Anxiety%20%282%29.R?dl=0>.

Reviewer 1: Regarding the analysis pipeline, the central piece of information is missing: How will the association between SVO and attachment style be tested? Will the authors use the same approach as van Lange et al. (e.g., MANOVA, controlling for gender) or a different one? Similarly, will SVO be determined in the same way as by van Lange et al. (e.g., removing not classifiable participants)? If not, what is changed and why?

Authors' Reply: If participants could not be classified, we also planned to remove them. For our final analyses, we indeed had planned to analyse the data both with and without sex as covariate (giving the original report the maximum chance to replicate). The main analyses were planned by contrasting prosocials with proselfs, as these were the main analyses in the original paper. We had also planned to run a number of auxiliary analyses to check the robustness of our results:

- run analyses with interaction year as predictor, to ensure that studies do not vary per year (as we also had some different measures throughout).
- we planned to run all analyses within each year.

Reviewer 1: * The described experiment closely replicates van Lange et al., except from different measures of attachment. According to the authors, these measures are psychometrically more sound (i.e., reliable, valid) than those used by van Lange et al.

* The authors convincingly describe that van Lange et al. used a poor measure of attachment, which might have produced spurious associations with SVO. This clearly indicates the need for a replication study.

* The original study already rested on a large sample and the authors' sample even exceeds this number. The sensitivity analysis is convincing and the required analyses should be sufficiently powered.

Authors' Reply: We thank the reviewer for her/his positive words.

Reviewer 1: * It is not clear whether the authors expect van Lange et al.'s study to replicate or not.

* Otherwise, the authors' assumptions seem plausible to me.

Authors' Reply: We don't think that, for the replication, it is relevant whether we had expected the result to replicate or not. We initially thought that the hypothesis was relatively reasonable, but that perhaps the relationship between SVO and attachment is slightly more contextualized (e.g., perhaps securely

attached individuals are prosocial in an environment that is friendly and not harsh, and are not prosocial in a harsh environment). However, because of the methods used in the original paper, we became increasingly more sceptical of the results.

Reviewer 1: * I have one major issue with this research: While the authors focus on better measures of attachment, they seem to have missed crucial developments regarding the assessment of SVO (see Murphy, Ackermann, & Handgraaf, 2011; Murphy & Ackermann, 2013). The scale used by van Lange et al. has meanwhile been shown to be clearly dominated by continuous SVO scales, and the use of nominal categories is commonly discouraged for theoretical reasons (e.g., Fiedler, Glöckner, Nicklisch, & Dickert, 2013). This said, I want to stress that I like the authors' attempt to test whether an important empirical finding still holds when improved measures are used, rather than literally repeating the same experiment. However, this approach would have required to measure SVO in a modern and empirically sound way as well. This aspect reduces my enthusiasm about the described experiment. Since the experiment has already been conducted, the authors should at least address this limitation later in the discussion of their manuscript.

Authors' Reply: We understand the reviewer's concern and our attempt is partly constrained by the data we had available (after all, we also started the data collection for this project in 2011 and ended in 2014; for the 2014, we could have adjusted the measure). We would like to point that, unlike many other SVO papers, we do calculate a Kuder-Richardson 20 coefficient for its reliability. If this coefficient is sufficient, we can be sufficiently confident that the measure does what it is supposed to do. We do agree that we will have to discuss the superiority of the new measure in the discussion.

Reviewer 1: * To re-iterate, I feel that the analysis pipeline is not described in sufficient detail to determine whether or not it is sound.

Authors' Reply: We hope that with the links to the syntaxes and the extra explanation this is sufficiently detailed.

Reviewer 1: * It would have desirable to have van Lange et al.'s original instruments in these experiments as well. This way, the authors could have tested whether differences in the findings reflect differences in the applied measures or result from other differences in the experimental protocol.

Authors' Reply: We understand this preference, but suspect that this may have been overkill. After all, it seems that the original results were obtained by

controlling for gender and dropping a few items. Given also the limited evidential value of the original article, the effect was likely noise.

Reviewer 2: This is a sound replication of previous research by Van Lange et al. (1997) on social value orientation and adult attachment. I have two minor suggestions that I think would help improve the paper. First, the authors seem to be implying that the large number of citations that the original article received is a marker of the paper's influence. But my sense, from examining some of the articles that have cited the original is that those citations are not for the attachment-SVO findings/claims, but rather simply a citation for the SVO idea itself and some of the methods that were used to study it. I think it is potentially misleading to imply that the attachment-SVO findings have been influential in the field.

Authors' Reply: We thank the reviewer for this comment, but respectfully disagree, for two reasons. First of all, the link with attachment provides the SVO measure with a proof-of-concept and if this proof-of-concept does not hold up, it potentially presents a problem for the measure. This impression is further confirmed by Van Lange et al.'s preference for conceptually replicating the effect. Furthermore, we looked at some of the most highly cited papers citing this paper (including Van Lange himself) and this further confirmed our idea:

- “Research regarding attachment processes also illuminates our understanding of dependence situations, in that issues of dependence and security are at the heart of attachment concerns. The adult attachment literature suggests that the intrapersonal and interpersonal adaptations acquired in childhood are carried into adult interactions.” (Rusbult & Van Lange, 2003)
- “As demonstrated in several studies, commitment is predictive of various cognitions and behaviors, including not only accommodation but also willingness to sacrifice, unrealistically positive beliefs about the relationship, and a shift in thinking from “I, me, and mine” to “we, us, and ours” (e.g., Agnew et al., 1998; Rusbult, Van Lange, Wildschut, Yovetich, & Verette, 2000; Van Lange, Otten, De Bruin, & Joireman, 1997; Van Lange, Rusbult, et al., 1997; Wieselquist, Rusbult, Foster, & Agnew, 1999).” (from Karremans, Van Lange, Ouwerkerk, Kluwer, 2003).
- “Another line of research suggests that SVO might be related to perceived closeness because of its relationship with attachment styles (Van Lange, De Bruin, Otten, & Joireman, 1997)” (from Cornellisen et al., 2011).
- “‘Cooperation’ in this literature is broadly defined and encompasses the contribution of points (e.g. De Cremer and van Vugt, 1999) or time

- (e.g. McClintock and Allison, 1989) in social dilemmas, pro-environmental behaviour (e.g. van Vugt et al., 1995), a friendly and cooperative negotiation style (e.g. De Dreu and van Lange, 1995; Nauta et al., 2002), and identification in close relations (e.g. van Lange et al., 1997a)” (from Bogaert et al., 2012).
- “Adults who are higher in attachment-related anxiety are less skilled at decoding nonverbal messages (Feeney, Noller, & Callan, 1994), display less "topical" reciprocity (a sign of responsive listening) in response to another's self-disclosure (Mikulincer & Nachshon, 1991), and are less likely to hold a prosocial interpersonal orientation (Van Lange, Otten, De Bruin, & Joireman, 1997)” (from Collins & Feeney, 2000).
 - “Some of the confusion about the difference between people’s personal value systems and their perceptions of others’ value priorities—people’s social value systems—stems from the tendency to discuss people’s stable tendencies to deal with others in their social environments in particular ways as social values (e.g., Beggan & Allison, 1994; Liebrand & Dehue, 1996; McClintock, 1978; Rokeach, 1973; Van Lange, Otten, De Bruin, & Joireman, 1997)” (from Rohan, 2000).
 - “A third avenue for future research involves an examination of how individual differences influence the extent to which status motives lead to self-sacrifice (e.g., Campbell, Simpson, Stewart, & Manning, 2003; Kurzban & Houser, 2005; Van Lange, Bekkers, Schuyt, & Van Vugt, 2007; Van Lange, Otten, De Bruin, & Joireman, 1997)” (from Griskevicius et al., 2010).
 - “Social motives may be rooted, for example, in individual differences in social value orientations (e.g., McClintock & Liebrand, 1988; Van Lange, Otten, De Bruin, & Joireman, 1997)” (from De Dreu et al., 2000).
 - “In addition to signaling wealth, public philanthropy can also convey prosocial personality traits (Penner & Finkelstein, 1998; Van Lange, Otten, DeBruin, & Joireman, 1997)” (from Griskevicius et al., 2007).
 - “Although theoretical models of group selection (3, 4), indirect reciprocity (5–7), and costly signaling (8) have helped to clarify possible evolutionary routes to cooperation, fundamental questions remain about the number and nature of the cognitive mechanisms that underpin human cooperative psychology (9) and whether there are stable individual differences in these mechanisms (10).” (10 refers to the Van Lange et al., 1997 article; from Kurzban & Houser, 2004).

Reviewer 2: Second, on page 5, I would caution against suggesting that adult attachment patterns are a function of "early" experience. The statement implicitly suggests that later experiences are not relevant. I doubt the authors

believe that, but that is one way to interpret the use of the term "early" rather than, say, "prior" or "a history of interpersonal experiences."

Authors' Reply: We agree with the reviewer and changed early to prior. We did retain "early" in the quote to Van Lange et al.'s work, as it was simply a quote to their work. Later, when referencing Van Lange et al.'s work, we include both early and prior.

We hope that by addressing the above concerns, the manuscript is ready for acceptance for Stage 1 review. If you have any other questions, please do not hesitate to contact us.

Sincerely,

Hans IJzerman
Jaap Denissen

Appendix C

Dear Dr. Chambers,

We would like to thank you again for your and the reviewers' thorough attention to our work. As requested, we registered our Stage 1 manuscript in the Open Science Framework (<https://osf.io/wxgv8/>). In addition, we submit a manuscript with track changes on and one with track changes accepted.

We hope that by doing so, the manuscript meets the standards for Stage 2 acceptance. If you have any other questions, please do not hesitate to contact us.

Sincerely,

Hans IJzerman
Jaap Denissen

Appendix D

Dear Prof Chambers, dear Chris,

We thank you and the reviewers for your helpful comments. Below we indicate how we responded to the reviewers' concerns:

Reviewer 1: The authors should explicate the number of individuals that could not be classified. Also there is currently no description of the criterion/cutoff used for the classification.

Authors' Response: We have included this information under our Confirmatory Results: "Our sample included 499 prosocials (65%) and 269 proselves (35%) and 95 individuals who could not be classified. If participants answered more than 6 dilemmas in a prosocial manner, they were classified as being prosocial"

Reviewer 1: Following my request from the first stage review, the authors have added a footnote on the availability of better SVO measures. However, I do not think that this should be put in a footnote, it deserves to be part of the main body of the discussion. First, the very philosophy of the authors (which I endorse!) is that replication should make use of the development of better measures. To be consistent with this thinking, the authors should not only stress to what extent their paper meets this standard but also point to their failures of meeting it (although it was clearly not a fault as new SVO instruments were freshly developed when data collection commenced). This does not weaken the paper but instead makes it more consistent and convincing. Second, the authors' argument that their SVO scale was highly consistent based on KR20 is at least difficult. On the one hand, this statistic rests on an artificial dichotomization of participants' answers. On the other hand, it is not surprising to get a high KR20s given that the triple dominance scales essentially confront participants with three highly similar choices (Murphy & Ackermann, 2014, PERS SOC PSYCHOL REV). Third (and related), the very point about newly developed SVO scales (e.g., Murphy, Ackermann, & Handgraaf, 2011, JUDGM DECIS MAK) is the insight that gradual differences in SVO can matter much more than nominal classifications - classifications can even conceal meaningful differences due to SVO (for illustrations of this important point, see Bieleke, Gollwitzer, Oettingen, & Fischbacher, 2017, J BEHAV DECIS MAKING; Fiedler, Glöckner, Nicklisch, & Dickert, 2013, ORGAN BEHAV HUM DEC). To me, this insight seems at least as important as the new developments regarding attachment scales stressed in the present paper. Addressing this point more thoroughly would make the paper a much more useful contribution for researchers who care about SVO. Moreover, it provides a promising avenue for future research - which, for instance, might examine whether gradual SVO differences relate more robustly to attachment.

Authors' Reponse: We have included the following information now in the discussion and are more explicit about it being a shortcoming in our research:

Do note however that in 2011, a psychometrically superior measure – the slider measure – for SVO became available (Murphy, Ackermann, & Handgraaf, 2011). Our data collection ran from 2011-2014, and the old, measure, with nominal categories, was still included in our test batteries (despite the superior measure having been available) and this was a shortcoming of our research. Because the slider measure has been shown to be superior to our measure (see e.g., Fiedler, Glöckner, Nicklisch, & Dickert, 2013), whether or not the slider measure does correlate with attachment is question for future research.

We again include a version with track changes on and track changes accepted to facilitate review. We hope that with these changes the manuscript meets the standards to be accepted for publication in Royal Society Open Science.

Sincerely and also on behalf of Jaap Denissen,
Hans IJzerman